# The Relationship between Cancer and Dementia: An Updated Review

**DOI:** 10.3390/cancers15030640

**Published:** 2023-01-19

**Authors:** Yung-Shuo Kao, Cheng-Chang Yeh, Yi-Fang Chen

**Affiliations:** 1Dr Kao Clinic, Taichung 406, Taiwan; 2Department of Oral Hygiene Care, Ching-Kuo Institute of Management and Health, Keelung 203, Taiwan; 3Department of Senior Citizen Service Management, National Taichung University of Science and Technology, Taichung 403, Taiwan

**Keywords:** cancer, cognitive impairment, dementia, cancer treatment

## Abstract

**Simple Summary:**

As cancer treatments continue to improve, more patients are surviving cancer. However, treatment-related complications, such as cognitive impairment, are becoming increasingly prevalent. Recent research suggests a link between cancer and dementia. Progress has been made in understanding the cognitive changes associated with cancer treatment, but more needs to be carried out to prevent and screen early for dementia in cancer survivors. In this review, we summarize the current literature on the incidence and mechanisms of cognitive impairment in patients with various types of cancer, including lung, breast, head and neck, gastric, prostate, colorectal, and brain tumors/metastases following different treatments. We also discuss potential risk factors and cellular mechanisms of neurodegeneration caused by cancer treatment in order to identify the early onset of cognitive deficits in cancer patients.

**Abstract:**

The risk of cancer and dementia increases with age, raising complex questions about whether it is appropriate to continue cancer treatment in older patients. There is emerging research suggesting the association between cancer and dementia. However, the mechanistic underpinnings are still under investigation. Progress has already been made toward understanding the cognitive effects associated with cancer therapy. Such associations raise awareness about the need to establish better prevention methods and early screening in clinical practice. Additionally, recent studies have suggested possible therapeutic strategies for better preserving cognitive function and reducing the risk for dementia before patients start cancer treatment. We review the current literature and summarize the incidence and mechanisms of cognitive impairment in patients with lung cancer, breast cancer, head and neck cancer, gastric cancer, prostate cancer, colorectal cancer, and brain tumor/brain metastasis following different kinds of therapies. Possible risk factors are suggested to identify the early onset of cognitive changes in cancer patients and provide more insight into the pathophysiological process of dementia.

## 1. Introduction

Cancer is now a leading health problem around the world. All kinds of cancers involve cellular transformation in morphology, dysregulation of apoptosis, unconstrained cellular proliferation, angiogenesis, invasion, and metastasis, accordingly, promoting tumor development [1]. The treatment strategy of cancer can involve surgery, chemotherapy, radiotherapy, target therapy, immunotherapy, and so on. With advances in early diagnosis and the improvement of aggressive treatments, a growing number of patients survive cancer; however, treatment-related complications are increasingly gaining attention. Many patients’ long-term side effects come from the cancer itself and occur months or years after treatment, such as early menopause, heart failure, or cognitive impairment. Long-term negative effects on cognitive function from cancer treatment may decrease the quality of life of patients and their families and increase the burden on the healthcare system. The incidence of cognitive impairment is multifactorial; although solid evidence is yet to be established, it is still worth investigating. While the detailed mechanism of cognitive decline is currently unclear, coagulative necrosis of brain tissue combined with fibrinoid changes to brain vessels, damage to the myelin sheaths of the neurons, and a decreasing number of glial cells have been found to cause axonal swelling and the destruction of the blood–brain barrier, which have been attributed to radiation-associated cognitive decline [2].

Cognitive decline can range from mild cognitive impairment to severe conditions such as dementia. Dementia is typically diagnosed based on acquired cognitive impairment with a severe influence on thinking and day-to-day function. The development of this disorder is progressive, meaning it starts out slowly and then eventually affects major life activities and limits social interaction. Due to the improvement in treatment outcomes in cancer patients and longer lifespans, the incidence of dementia has increased significantly. In Western countries, the prevalence is 5–20% in people aged over 65, and higher among women than men [3]. Alzheimer’s disease (AD) is the most common type of dementia, responsible for 70–80% of all cases [4]. Other common types of dementia include vascular dementia, Lewy body dementia (LBD), frontotemporal dementia (FTD), and Parkinson’s disease dementia (PDD). As of yet, there is no effective treatment that can cure Alzheimer’s disease and alter the disease process in the brain. In late stages, complications from the destruction of brain function will result in death. Dementia is related to many diseases, such as diabetes and non-alcoholic fatty liver disease [5,6]. There is more and more evidence indicating a direct link between cancer and dementia. The positive relationship between cancer and dementia is found to be involved in the pathogenesis of both cancer and dementia, although the exact link and mechanistic underpinnings remain unclear. Nevertheless, some studies have revealed an inverse association that showed that the risk of dementia in cancer patients after chemotherapy is lower [7]. A substantial amount of bias may have an influence on the direction of the association, which makes the relationship more complicated.

Over the past few decades, clinical oncology has made great advances that have significantly improved long-term survival for cancer patients. The success of cancer treatment means that more cancer survivors may experience different sequelae lasting months or years after treatment. In the increasing patient population of multiple cancers, cognitive impairment will be a serious issue requiring intensive research to advance the development of early diagnosis and appropriate prevention and management. The discovery of risk factors of cognitive impairment is urgently needed for the early identification of cancer survivors at a high risk of dementia and the development of strategies to reduce the burden on medical care in an aging society.

We use the diagram below to indicate the risk factors and mechanisms of neurodegeneration from cancer treatment (Figure 1).

## 2. Lung Cancer and Dementia

Lung cancer is the leading cause of cancer-specific mortality; the five-year survival rates for advanced lung cancer vary from 2.6% to 23.2% [8,9]. More than half of patients diagnosed with lung cancer are in the advanced stage of the disease. Generally, about 84% of lung cancer pathological types are non-small-cell lung cancer (NSCLC) and about 13% are small cell lung cancer (SCLC) [10]. Depending on the stage of lung cancer, patients receive treatments including surgery, radiation, chemotherapy, and targeted therapy.

With advances in aggressive treatments, a growing number of patients survive cancer; however, treatment-related complications are gaining increasing attention [11]. Long-term effects on brain function resulting from lung cancer treatment are an important complication that negatively affects the quality of life of cancer patients. For years, research has focused on chemotherapy as a driver of brain dysfunction—some cancer survivors refer to this phenomenon as “chemo brain”. More and more studies have identified different mechanisms of chemotherapy-induced cognitive issues, such as oxidative damage, DNA damage, telomere shortening, hormonal change, and neurotoxicity, all of which contribute to the development of cognitive decline in patients after chemotherapy [12]. Regarding previous neuroimaging results, strong support has been provided for acute structural and functional changes in brain regions undergoing chemotherapy treatment [13,14].

Epidermal growth factor receptor targeting tyrosine kinase inhibitors (EGFR-TKI) are the standard treatment for NSCLC patients with EGFR mutations worldwide. EGFR has been previously demonstrated to act as a signaling nexus for many biological pathways involved in the proliferation, regeneration, and development of neurons [15]. The inhibition of EGFR signaling may have a negative impact on neurodevelopment and cause severe functional decline that is associated with the development of cognitive impairments. A study found an increase in white matter lesions and decrease in grey matter volume in metastatic NSCLC patients following long-term EGFR-TKI treatment, which indicated that structural and functional changes in the brain are strongly correlated with the cognitive status of patients [16]. However, one pilot study that compared the effects of targeted therapy and chemotherapy on neuropsychological performance in patients with non-small-cell lung cancer showed no significant difference in terms of abnormal cognitive performance [17]. Although more studies have focused on systemic lung cancer treatment, e.g., chemotherapy, growing evidence suggests that cognitive impairment may occur before, and independently of, cancer treatment [18]. Accumulating data have implied that additional factors, such as elevated levels of proinflammatory cytokines, are implicated in the pathogenesis of neurological and cognitive dysfunction. Recent research that assessed the frequencies of neuronal autoantibodies in patients with lung cancer and indicated their association with cognitive decline has provided a potential mechanism of immune-mediated cognitive dysfunction [19]. Among NSCLC patients, patients with immunoglobulin A autoantibodies that target the N-methyl-d-aspartate receptor had a higher risk of developing deficits in verbal memory.

The radiotherapy issue in lung cancer is complicated. Prophylactic cranial irradiation (PCI) is frequently used in the management of small cell lung cancer. PCI is associated with cognitive function decline. A recent study, NRG-CC003, tried to use the hippocampal avoidance whole-brain irradiation technique; this study is still ongoing [20]. The reader could refer to the section “Brain tumor/Brain metastasis and dementia” for more details about the hippocampal avoidance whole-brain irradiation technique.

## 3. Breast Cancer and Dementia

Female breast cancer is the most common malignancy in the world. Apart from sex and age, a decline in physiological function is the most significant risk factor for breast cancer development [21]. Treatments for patients with breast cancer have historically consisted of surgery, chemotherapy, hormonal therapy, and radiation therapy, depending on the type and stage of breast cancer and how far it has spread. With advances in early diagnosis and the improvement of aggressive treatments, a growing number of patients survive breast cancer; however, treatment-related complications remain a critical concern.

Chemotherapy has a general goal of killing cells that are growing or dividing; however, treatment can destroy the central nervous system. Cognitive decline during or after chemotherapy has occurred in many patients, and reported deficits, such as difficulty with attention, concentration, planning, and working memory, have been found in 17% to 75% of breast cancer survivors in research conducted from six months to even 20 years following chemotherapy [22]. The mechanisms of chemotherapy-induced cognitive problems in breast cancer survivors were previously revealed to include brain aging acceleration and neurotoxicity affecting the central and peripheral nervous system. Age-related molecular markers are increasingly expressed and gray matter atrophy, such as brain aging, is shown after breast cancer patients receive chemotherapy [23,24,25]. These studies provided support for measuring structural brain networks and detecting neuro-images and biomarkers that may improve the accurate early prediction of AD in breast cancer survivors. In addition, the genetic risk for breast cancer, the APOE ε4 genotype, is associated with the increased expression of amyloid ß (Aß) and tau may trigger the neuropathology of AD. Both Aß and tau are expressed by malignant breast tumor cells and breast cancer chemotherapy harms autophagy, which may also induce the accumulation of Aß and tau [26]. Although previous studies reported the presence of a ε4 allele that positively mediated the incidence of breast cancer, the currently available literature on associations between APOE polymorphism and breast cancer and breast cancer chemotherapy risk is inconsistent. Moysich et al. found no significant association between the ε4 allele variant and malignant breast tumors [27]. Another longitudinal study showed that breast cancer survivors with the APOE ε4 allele had no significant worsening of cognitive function over time and no vulnerability to chemotherapy exposure [28]. As APOE ε4 is only present in a small percentage of the population, the low sample size of APOE ε4 breast cancer patients may result in differences and limitations, and the substantial selection bias would also influence the direction of the association. Further studies will need to be conducted to investigate the role of the biomarkers of neurodegeneration to evaluate the risk of chemotherapy-related cognitive decline. In other observation studies, chemotherapy was not associated with a greater risk of dementia in older breast cancer survivors [29,30]. However, very elderly women (75 or older) at a higher risk of developing dementia may have less opportunity to receive chemotherapy and people with comorbidities, such as diabetes mellitus and hypertension, would have an increased probability of dementia being diagnosed. Age and comorbid conditions are factors influencing the delivery of chemotherapy; thus, these findings may not be able to be extrapolated to younger patients. In addition, whether Medicare claims data could reflect the subtle cognitive effect of chemotherapy in identifying patients with dementia, and whether the follow-up time is sufficient to progress to dementia being diagnosed, are matters that will need to be considered.

Adjuvant hormonal therapy, such as tamoxifen and aromatase inhibitors, significantly improves the long-term survival of women with hormone receptor-positive breast cancer. Previous studies have suggested that hormonal therapy lowers the expression of estrogen and/or progesterone in the body and blocks their effects on breast cancer cells. Based on the long-term effects of hormone therapy, a significant improvement of survival after breast cancer was reported over time and estrogen could suppress the cell-death pathway, therefore contributing to a neuroprotective effect on cognitive impairment. According to multiple surveys, among patients with breast cancer, treatment with tamoxifen and aromatase inhibitors was associated with a decrease in the incidence of AD and dementia [31,32]. The combined use of tamoxifen and aromatase inhibitors can significantly reduce the risk of dementia in breast cancer patients [33]. However, other clinical studies have yielded inconsistent results regarding tamoxifen treatment and dementia. One Danish study indicated no clinically relevant association between the use of tamoxifen or other endocrine therapies and the risk of dementia [34]. The SEER-Medicare database found no association between the definitive endocrine therapy treatment of nonmetastatic breast cancer and the risk of dementia, which contradicts some prior findings showing a protective effect [35]. Possibly due to differences in patient numbers between studies and selection bias, dementia diagnosis from the registry rather than a neuropsychological assessment would be taken into account. In evaluating the effect of endocrine therapies on cognitive function, a lack of menopausal status information might result in inaccurate findings and lead to a failure to determine the effects of endocrine therapy on cognitive function.

Radiotherapy for breast cancer is administered to reduce the likelihood of tumor recurrence and is the standard treatment following surgical removal in some breast cancer patients; however, its long-term sequelae remain imperfect. Some breast cancer patients with long-term survival might experience unexpected late side effects. Previous research has shown that breast cancer patients receiving adjuvant regional radiation may have cognitive impairment even several months after radiotherapy [36]. Among the clinical complications reported, even when such therapy does not directly apply to brain areas, protracted cognitive impairment has been observed to occur after radiotherapy. The radiation dose to the whole breast in the study is 5000 centi-Gray (cGy), given in 25 fractions using the tangential technique [36]. The recent American guideline on whole-breast irradiation is to use hypofractionation radiotherapy, either 4000 cGy in 15 fractions or 4256 cGy in 16 fractions [37]. Some of the United Kingdom’s radiation oncologists use an ultra-hypofractionated regimen, 2600 cGy in five fractions over one week [38]. Future research should focus on the relationship between hypofractionated radiotherapy and dementia in breast cancer patients.

## 4. Head and Neck Cancer and Dementia

Head and neck cancer is prevalent around the world. The major types of head and neck cancer include nasopharyngeal cancer, oral cavity cancer, oropharyngeal cancer, hypopharyngeal cancer, and laryngeal cancer. The standard treatment for nasopharyngeal cancer is radiation-based therapy, while the standard treatment for oral cavity cancer is surgery, radiotherapy, and chemotherapy may be given as an adjuvant therapy, according to the pathology [39]. The main treatment for oropharyngeal cancer, hypopharyngeal cancer, and laryngeal cancer can be surgery or organ preservation, depending on the condition and willingness of the patient. Sinonasal cancer is rare, but the origin of the disease is close to the brain, thus we need to mention it in this section.

The radiation therapy dose for head and neck malignancy often influences the lower brain, which can cause dementia, as previous studies showed [40,41,42,43]. The quality of life in head and neck cancer patients with dementia is decreased [44]. As a result, there have been many attempts to lower the radiation dose delivered to the brain. For example, proton therapy has become popular in recent years. The Bragg peak property of the proton beam can better spare normal tissue, thus reducing subsequent side effects. According to a previous report, grade 3 or higher temporal lobe necrosis is about 0.9% in head and neck cancer patients receiving proton therapy to the skull base [45]. Another strategy is the use of carbon ion therapy. Carbon ion beams also exhibit Bragg peak properties as proton beams. The major difference between carbon ion beams and proton ion beams is the relative biological effectiveness (RBE). The RBE is defined as the ratio of doses to reach the same biological effect when comparing a 250 KVp X-ray to the target beam [46]. The RBE of proton beams is about 1.1 [46]. The RBE of carbon beams is about 3. Because the RBE of carbon beams is higher than that of proton beams, carbon ions could deliver more damage to the tumor than proton beams in the same condition. A recent study showed that carbon ion therapy can deliver a lower dose to the temporal lobe [47].

Another problem in radiotherapy is re-irradiation. Since head and neck cancer patients now live longer due to advances in cancer treatment, it is natural that the probability of retreatment may rise. The retreatment of nasopharyngeal cancer is often by surgery; however, if surgery is not feasible, re-irradiation should be considered. However, re-irradiation will cause temporal lobe necrosis. According to a meta-analysis, the pooled grade 3 or higher temporal lobe necrosis is 19% [48]. Thus, the monitoring of cognitive function is important in the re-irradiation of nasopharyngeal cancer patient groups.

A meta-analysis demonstrated that cognitive decline following radiation therapy in head and neck cancer patients was strongly associated with structural and functional changes in magnetic resonance imaging (MRI) [49]. This finding indicates that we can use MRI to track the cognitive function in this patient group.

## 5. Gastric Cancer and Dementia

Gastric cancer is a common cause of death in both the West and the developing world. The main form of treatment of gastric cancer is surgery, which may be followed by chemotherapy or not, according to the surgical pathology. Lauren’s classification is a commonly used histological classification in gastric cancer. According to Lauren’s classification, there are two common types of histology, intestinal type and diffuse type. Lauren’s classification can further be used to personalize the chemotherapy regimen and serve as a prognostic factor [50,51].

A study conducted in Korea showed that gastrectomy increased the risk of Alzheimer’s disease in gastric cancer patients, but patients who received a vitamin B12 supplement had a decreased risk of developing Alzheimer’s disease [52]. Vitamin B12, also called Cobalamin, is an indispensable water-soluble nutrient [53]. A vitamin B12 deficiency has been shown to be related to dementia [54]. The physiology of vitamin B12 is complicated. Vitamin B12 is liberated from binding proteins in the stomach and then absorbed in the intestine [55]. As a result, gastrectomy will destroy the vitamin B12 absorption pathway. As a result, supplements of vitamin B12 are important in patients receiving a gastrectomy.

## 6. Prostate Cancer and Dementia

Prostate cancer is the main cancer in men. The treatment of prostate cancer may involve surgery, anti-androgen therapy (ADT), chemotherapy, and radiotherapy. The clinical algorithm of prostate cancer is strongly dependent on life expectancy and prognostic groups. If the life expectancy is short, then observation or active surveillance can be considered.

ADT is often used as the frontline treatment in the management of prostate cancer. The goal of ADT is to lower the testosterone level in patients. According to NCCN guidelines, the treatment time of ADT was determined by the risk group of prostate cancer [56]. ADT was shown to be related to an increased risk of dementia [57]. Orchiectomy, luteinizing hormone-releasing hormone agonists, oral androgen antagonists, and combined androgen blockade increased the subsequent risk of dementia [57]. The detailed mechanism of the correlation between ADT and dementia is still under investigation. The side effects of ADT, including the loss of lean body mass, diabetes mellitus, cardiovascular disease, and depression, are risk factors for Alzheimer’s disease and dementia [58]. This association may explain the relationship between ADT and dementia. Another possible mechanism is that lowered plasma levels of testosterone are associated with an increased plasma concentration of amyloid β [59]. For dementia prevention, statins can lower the risk of dementia in prostate cancer patients with diabetes receiving ADT [60].

Dementia diagnosis in prostate cancer patients has a negative effect on survival [61]. As a result, the use of ADT should be based on updated scientific evidence and sharing the decision-making with the patient and their family is important in the clinical setting.

## 7. Colorectal Cancer and Dementia

Colorectal cancer is a term containing two entities, colon cancer and rectal cancer. The treatment strategies for colon cancer and rectal cancer are quite different. The main treatment for colon cancer involves surgery, then subsequent chemotherapy may be given or not, as determined by the surgical pathology. The current standard treatment of rectal cancer is neoadjuvant radiotherapy with chemotherapy, followed by surgery.

There are many studies investigating the relationship between colorectal cancer and dementia. A study using a Taiwanese database showed that chemotherapy increased the risk of dementia in colorectal patients older than 80 [62]. Another study using the SEER database showed that chemotherapy could reduce the risk of dementia when compared with non chemotherapy users in colorectal cancer [63]. It seems that we need more evidence to understand the relationship between colorectal cancer and subsequent dementia.

Dementia in colorectal cancer was associated with poor survival [64]. Additionally, a previous study had shown that chemotherapy may increase survival in dementia patients [65].

Another issue is colon cancer screening in dementia patients. Alzheimer’s disease and related dementia were associated with a lower usage rate of fecal occult blood tests, colonoscopies, and sigmoidoscopies [66]. This public health issue needs to be addressed to promote colon cancer screenings in dementia-prone patient groups.

## 8. Brain Tumors/Metastases and Dementia

Primary brain tumors mean that the tumor initially developed in the brain. There are many types of brain tumors, such as glioblastoma multiforme, medulloblastoma, ependymoma, CNS lymphoma, etc. The treatment of primary brain tumors varies significantly. Glioblastoma multiforme, medulloblastoma, and ependymoma are initially treated with surgery, followed by radiation and chemotherapy. The treatment of CNS lymphoma only involves radiotherapy and chemotherapy. It is natural to assume that brain tumors cause dementia due to the tumor distorting and destroying the brain. The REGI-MA-002015 trial showed that proton therapy is better at the preservation of cognitive functioning for intracranial tumors than traditional radiotherapy [67].

Brain metastasis means that the primary tumor’s origin is not in the brain, but the primary tumor later develops a metastasis in the brain. The treatment of brain metastasis is a fast-developing research area. The main treatment of brain metastasis is surgery, followed by radiotherapy [68]. However, if the brain metastasis is not operable, then radiotherapy is the main treatment strategy [68].

The most common radiotherapy technique for brain metastases is whole-brain radiotherapy (WBRT). However, whole-brain radiotherapy often causes a significant decline in cognitive function. As a result, many strategies have been developed for better preserving cognitive function. The hippocampus is the region of new memory formation, so lowering the radiation dose to the hippocampus can be beneficial for cognitive functioning. RTOG 0933, a phase II multi-institutional clinical trial, showed that hippocampal-avoidant WBRT can reduce cognitive function decline [69]. Another strategy is the use of memantine. RTOG 0614, a randomized controlled trial, showed that memantine could better preserve cognitive function in patients receiving WBRT [70]. The NRG CC001 trial combined memantine and hippocampal-avoidant WBRT, and the combination worked well [71].

Another strategy is stereotactic radiosurgery (SRS), which can concentrate the radiation dose to a small area, thus protecting cognitive function [72]. Traditional SRS is a single fraction with a very high dose, and thus can only be used for small brain tumors. Recently, fractionated SRS using a lower fraction size has become popular and is widely used for larger tumors [73]. Future data about dementia in patients with a larger tumor size can be anticipated.

## 9. Commonly Used Strategies to Reduce Dementia in Cancer Patients

Although the detailed mechanism of the association between cancer and dementia is still under investigation, some cancer treatments are strongly associated with a decline in cognitive function in cancer patients. As a result, there are some strategies that are commonly used to reduce dementia in cancer patients. We list them in the table below (Table 1). There are some studies investigating the age cut-off for PCI [74,75]. However, there are conflicts among the included studies. Whether the PCI should be avoided in the elderly patients is an unsolved clinical problem; nonetheless, we also present it in the table.

## 10. Summary of Current Evidence of the Association between Cancer and Dementia

We summarized the current evidence of the association between cancer and dementia in Table 2, and that can be applied in the clinical setting. As we can see in Table 2, the surgery and the target therapy in cancer patients and the risk of dementia were rarely investigated. The future direction should focus on these two treatment modalities.

## 11. Cellular Mechanisms of Cancer-Related Effects on Neurodegeneration

The cellular mechanisms of cancer-related effects on neurodegeneration are summarized in Figure 2. Despite the complex relationship between cancer and dementia, there are several promising cellular regulation mechanisms that can be targeted. Peripheral or central inflammation, DNA damage, the activation of apoptosis, the suppression of neurogenesis and gliogenesis, neurotransmitters dysregulation and reduction, increasing phosphorylation of tau protein, a reduced level of estrogen/ testosterone and their effects on the brain, the inhibition of cerebral blood flow, and the reduction in the number of cortical spines and dendrites are possible mechanisms involved during and after cancer treatment [76,77,78,79,80,81,82]. With knowledge of the consequences of cancer therapy and the underlying mechanisms, progress would be made regarding treat-to-target and less neurotoxin strategies for cancer survivors. Eventually, with the combination of the development of appropriate and early prevention in clinical and public health practice, the cognition capacity of cancer patients would be restored and disease burdens of both cancer and dementia would be reduced.

## 12. Factors Predispose Patients for Dementia following Cancer Therapy

Currently evidence regarding the role of genetic factors, morbidity, and biomarkers in the development of dementia following cancer therapy is summarized in Table 3. The genetic factors are APOE-4, IL-1R1, COMT, BDNF, BIN1, TOMM40, and OPRM1 and they involve related mechanisms, including blood brain barrier dysfunction, oxidative stress, inflammation, neurotransmitter metabolism, neuroplasticity meditation, and tau pathology [83,84,85,86,87]. Diabetes and stroke are both risk factors for dementia. Diabetes can lead to damage in blood vessels, which contributes to the development of dementia [88]. Similarly, stroke can cause brain damage, which also increases the risk of dementia [89].

Correlations between cognitive impairment and inflammatory cytokine and receptors are currently being investigated. The associated biomarkers are TNF-α, TNF-receptor I/II, and interleukins (IL-1β, IL-2, IL-10, IL-6, IL-8). In addition, neurological markers, N-acetylaspartate/choline, N-acetylaspartate/myo-inositol, and serum axonal phosphorylated neurofilament heavy subunit H (pNF-H) are detected in patients undergoing cancer treatments [83,90,91,92], suggesting the potential applications as biomarkers in clinical practice.

## 13. Conclusions

With the increase in average life expectancy and chronic diseases, the prevalence of age-associated diseases, cancer, and dementia in the elderly population has risen rapidly. Aging populations require healthcare providers to develop advanced knowledge to meet the needs of the increasing number of cancer patients and dementia sufferers among the aging population. This review has summarized and clarified the underlying association between cancer and dementia among patients undergoing different forms of therapy and has included aging, cancer disease characteristics, genetic/social/psychological/lifestyle factors, cognitive reserve, and comorbidity, all of which contribute to the risk of cognitive decline. Moreover, the incidence of dementia is associated with chemotherapy, radiotherapy, hormonal therapy, and other systemic therapies. Possible mechanisms such as oxidative damage, DNA damage, radiation injury, deregulated immune/inflammatory response, inhibition of neurogenesis/vascularization, hormonal change, white/gray matter damage, and neurotoxicity are assumed to be responsible for the development of cognitive impairment. As no standard care strategies exist for clinical practice, further studies are needed to provide more insight into the association; in addition, clinicians and public health authorities should be aware of this important issue.

## Figures and Tables

**Figure 1 cancers-15-00640-f001:**
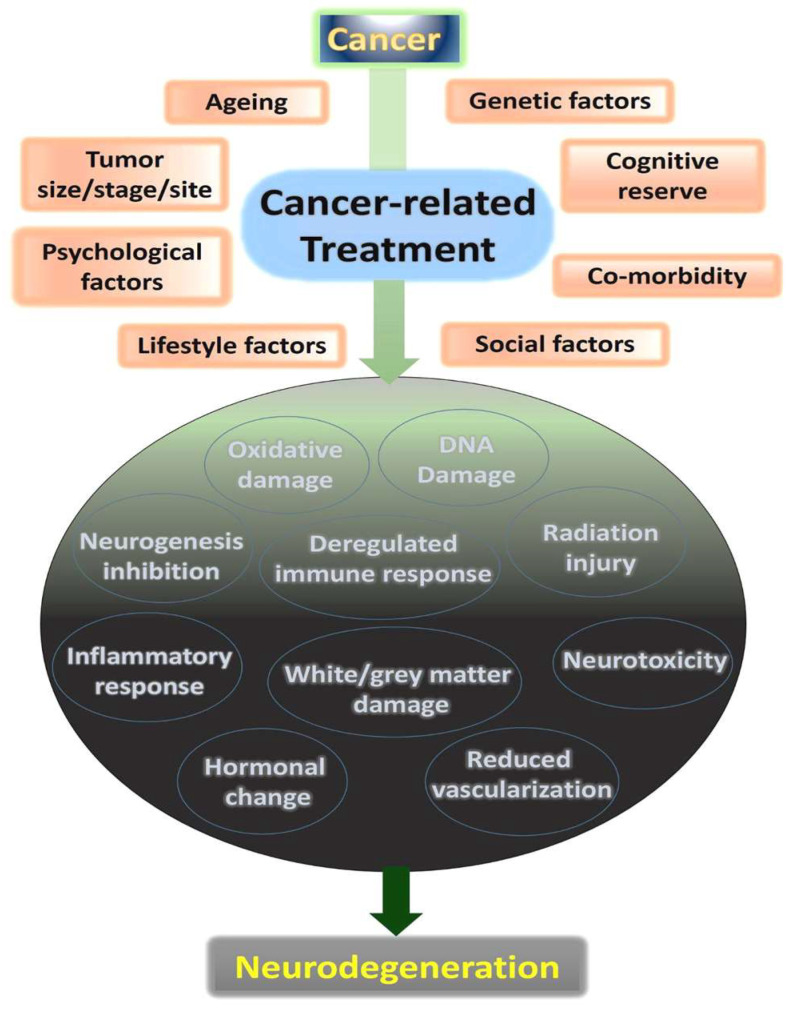
Risk factors and mechanisms of neurodegeneration from cancer treatment.

**Figure 2 cancers-15-00640-f002:**
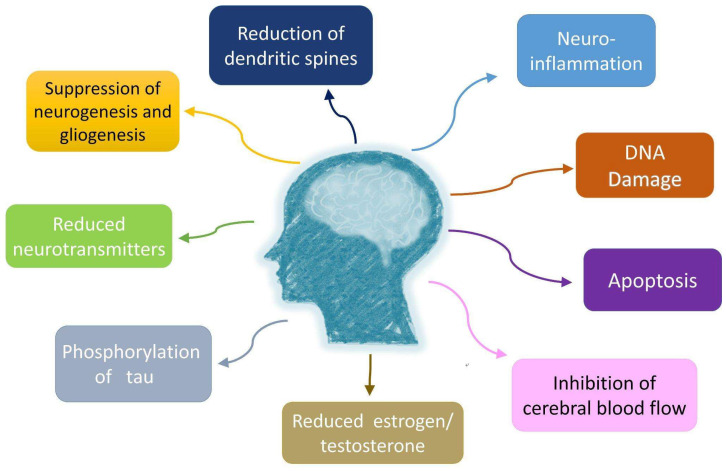
Several potential cellular mechanisms of cancer treatment inducing neurodegeneration have been proposed. Neuroinflammation, DNA damage, apoptosis, suppression of neurogenesis and gliogenesis, reduced neurotransmitters, phosphorylation of tau, reduced level of estrogen/ testosterone and their effects on brain, inhibition of cerebral blood flow, reduction of cortical spines, and dendrites are possible mechanisms involved.

**Table 1 cancers-15-00640-t001:** Commonly used strategies to reduce dementia in cancer patients.

Condition	Commonly Used Strategy
Brain metastasis	Hippocampal-avoidance whole-brain radiotherapy
Brain metastasis	Memantine for patients receiving whole-brain radiotherapy
Brain metastasis	Stereotactic radiosurgery
Brain tumor	Proton therapy
Head and neck cancer	Proton therapy
Head and neck cancer	Carbon ion therapy
Lung cancer	Avoidance of prophylactic cranial irradiation in elderly patients
Prostate cancer	Use of statins in diabetic patients with prostate cancer
Gastric cancer	Vitamin B12 supplement for patients receiving gastrectomy

**Table 2 cancers-15-00640-t002:** The association of cancer and dementia in the clinical setting.

Cancer Type	Surgery	Radiotherapy	Chemotherapy	Hormone Therapy	Target Therapy
Lung cancer	?	+	O	N/A	O
Breast cancer	?	+	O	+/−	?
Head and neck cancer	?	+	?	N/A	?
Gastric cancer	+	?	?	N/A	?
Prostate cancer	?	?	?	+	?
Colorectal cancer	?	?	+	N/A	?
Brain tumor/ brain metastasis	?	+	?	N/A	?

Abbreviation: ?: not reported; +: increase; +/−: inconsistent result; O: no effect; N/A: no use of this kind of therapy in the specific cancer type.

**Table 3 cancers-15-00640-t003:** Factors predisposing patients to dementia following cancer therapy.

		Possible Mechanism/Function
Genetic factors	APOE-4	Blood brain barrier dysfunction, oxidative stress, and inflammation
	IL-1R1	Inflammation
	COMT	Neurotransmitter metabolism
	BDNF	Mediation of neuroplasticity
	BIN1	tau pathology
	TOMM40	α-synuclein accumulation, oxidative damage, mitochondrial dysfunction and neuroinflammation
	OPRM1	Activation of the mu-receptor
Morbidity	Stroke	Brain atrophy/brain damage
	Diabetes	Induction of vascular damage
Biomarkers	pNF-H	Serum marker of axonal damage
	TNF-α, sTNFRI/II	Inflammatory cytokine and receptor
	IL-1β, IL-2, IL-10, IL-6, IL-8	Alteration of blood brain barrier
	NAA/Cho, NAA/mI	Neurological markers

APOE, apolipoprotein E; BDNF, brain-derived neurotrophic factor; COMT, catechol-O-methyltransferase; IL1-R1, interleukin-1-receptor1; BIN1, bridging integrator 1; OPRM1, mu opioid receptor gene; pNF-H, phosphorylated neurofilament heavy subunit; TNF-a, tumor necrosis factor-alpha; sTNFRI/II, tumor necrosis factor-receptor type I/II; Cho, choline; mI, myo-inositol; NAA, N-acetylaspartate; TOMM40, translocase of outer mitochondrial membrane 40.

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
