# Peer review of "The Relationship between Cancer and Dementia: An Updated Review"

_cancers, 2023, doi:10.3390/cancers15030640_

Round 1

Reviewer 1 Report

The authors describe the association between several cancers and dementia.

Creating a table and summarizing it would be helpful.

Author Response

Reviewer 1:

The authors describe the association between several cancers and dementia.

Creating a table and summarizing it would be helpful.

Reply: Thanks for your professional opinions! We have added table 2 and summarize the current evidence of the association between several cancers and dementia.

Reviewer 2 Report

Dear authors,

I had the opportunity to read your interesting manuscript.

The topic is of high clinical importance, reflected by the increasing number of publications in this area. Therefore it is mandatory to stress the uniqueness of your own review (what is different compared to the existing literature). Figure 1 is very helpful. I think you could/should ass another figure to illustrate the postulated changes on cellular level. Moreover, a more uniformly arrangement of the single cancer entities would be favourable.

Kind regards

Author Response

Reviewer 2:

Dear authors,

I had the opportunity to read your interesting manuscript.

The topic is of high clinical importance, reflected by the increasing number of publications in this area. Therefore it is mandatory to stress the uniqueness of your own review (what is different compared to the existing literature). Figure 1 is very helpful. I think you could/should ass another figure to illustrate the postulated changes on cellular level. Moreover, a more uniformly arrangement of the single cancer entities would be favourable.

Reply: Thanks for your professional opinions! This review is an updated review with the latest literatures, and we also present the results and association with dementia for each cancer type. Our review is unique.

Additionally, we had added figure 2 at the end of the article. Also, we had rearranged the discussion of single cancer entities in the manuscript.

Reviewer 3 Report

Introduction is too long should  be more closed to the argument.

Avoid doubling information issues in the introduction

Shrink rows 25-31

Shrink rows 53-76, be more close to the argument of the review.

Further long sub-introductions are  present in any sub-setting analyzed, this make the manuscript particularly baring.

Overall the elements in the review are quite foggy described. It is clear the absence of a specialist in oncology in the paper generating and writing, the description is indeed somewhat generic, without any practical conclusion and impact for daily clinical practice.   

In the manuscript there are very low indications about the assessment on cancer dementia and what we have to do to prevent it in the various subsetting, (eg age cut off for PCI etc), how to treat these patients and how to prevent its development in high risk individuals.

A discussion session is also imperative to analyze the problem and giving future projections.

Author Response

Reviewer 3:

Introduction is too long should be more closed to the argument.

Avoid doubling information issues in the introduction

Shrink rows 25-31

Shrink rows 53-76, be more close to the argument of the review.

Reply: Thanks for your professional opinions, we had shortened the introduction section.

Further long sub-introductions are present in any sub-setting analyzed, this make the manuscript particularly baring.

Reply: Thanks for your professional opinions, we had shortened the long sub-introductions.

Overall the elements in the review are quite foggy described. It is clear the absence of a specialist in oncology in the paper generating and writing, the description is indeed somewhat generic, without any practical conclusion and impact for daily clinical practice.

Reply: Thanks for your professional opinions. We had added practical conclusion and impact for daily clinical practice in the Table 2. Also, Dr. Yung-Shuo Kao is a radiation oncologist and serves as the cancer advisor in Dr. Kao Clinic. We had updated the affiliation in the manuscript.

In the manuscript there are very low indications about the assessment on cancer dementia and what we have to do to prevent it in the various subsetting, (eg age cut off for PCI etc), how to treat these patients and how to prevent its development in high risk individuals.

Reply: Thanks for your professional opinions! We had searched the related literatures and found that there are some studies investigating the age cut-off for PCI (Ref 74, 75). However, there are conflicts among the included studies. Whether the PCI should be avoided in the elderly patients in an unsolved clinical problem, we have added it in the section 9.

Ref 74. J Geriatr Oncol. 2015 Mar;6(2):119-26. doi: 10.1016/j.jgo.2014.11.002. Epub 2014 Dec 4. PMID: 25482023; PMCID: PMC5722214.

Ref 75. J Radiat Res. 2019 Oct 23;60(5):630-638. doi: 10.1093/jrr/rrz040. PMID: 31165148; PMCID: PMC6805975.

A discussion session is also imperative to analyze the problem and giving future projections.

Reply: Thanks for your professional opinions! We have added the discussion section in the section 9, 10, and 11. Section 9 summarized the commonly used strategies to reduce dementia in cancer patients. Section 10 summarized the current evidence of the association of cancer and dementia in the clinical setting. Section 11 summarized the cellular level mechanism of cancer-related effects on neurodegeneration. We also suggest the future direction in the section 9-12.

Round 2

Reviewer 2 Report

Well done revision!

Author Response

Reviewer 2: Well done revision!

Reply: Thank you for the kind words

Reviewer 3 Report

Overall the elements in the review are quite foggy described. The description is indeed somewhat generic, without any practical conclusion and impact for daily clinical practice.   

Author Response

Reviewer 3: Overall the elements in the review are quite foggy described. The description is indeed somewhat generic, without any practical conclusion and impact for daily clinical practice.  

Reply: Thanks for your professional opinions. We had summarized and added practical conclusion and impact for daily clinical practice in  Table 2 , 3 and section 12.
